# Association between On-Treatment Haemoglobin A_1c_ and All-Cause Mortality in Individuals with Type 2 Diabetes: Importance of Personalized Goals and Type of Anti-Hyperglycaemic Treatment

**DOI:** 10.3390/jcm9010246

**Published:** 2020-01-17

**Authors:** Emanuela Orsi, Enzo Bonora, Anna Solini, Cecilia Fondelli, Roberto Trevisan, Monica Vedovato, Franco Cavalot, Gianpaolo Zerbini, Susanna Morano, Antonio Nicolucci, Giuseppe Penno, Giuseppe Pugliese

**Affiliations:** 1Diabetes Unit, Fondazione IRCCS “Cà Granda—Ospedale Maggiore Policlinico”, 20122 Milan, Italy; emanuela_orsi@yahoo.it; 2Division of Endocrinology, Diabetes and Metabolism, University and Hospital Trust of Verona, 37126 Verona, Italy; enzo.bonora@univr.it; 3Department of Surgical, Medical, Molecular and Critical Area Pathology, University of Pisa, 56126 Pisa, Italy; anna.solini@med.unipi.it; 4Department of Medicine, Surgery and Neurosciences, Diabetes Unit, University of Siena, 53100 Siena, Italy; c.fondelli@ao-siena.toscana.it; 5Endocrinology and Diabetes Unit, Azienda Socio-Sanitaria Territoriale Papa Giovanni XXIII, 24127 Bergamo, Italy; rtrevisan@asst-pg23.it; 6Department of Clinical and Experimental Medicine, University of Padua, 35128 Padua, Italy; monica.vedovato@aopd.veneto.it; 7Metabolic Diseases and Diabetes Unit, Department of Clinical and Biological Sciences, University of Turin, 10043 Orbassano, Italy; franco.cavalot@unito.it; 8Complications of Diabetes Unit, Division of Metabolic and Cardiovascular Sciences, San Raffaele Scientific Institute, 00163 Milan, Italy; zerbini.gianpaolo@hsr.it; 9Department of Experimental Medicine, “La Sapienza” University, 00161 Rome, Italy; susanna.morano@uniroma1.it; 10Centre for Outcomes Research and Clinical Epidemiology (CORESEARCH), 65124 Pescara, Italy; nicolucci@coresearch.it; 11Department of Clinical and Experimental Medicine, University of Pisa, 56124 Pisa, Italy; pgiuse@immr.med.unipi.it; 12Department of Clinical and Molecular Medicine, “La Sapienza” University, Via di Grottarossa 1035-1039, 00189 Rome, Italy

**Keywords:** type 2 diabetes, HbA_1c_ all-cause mortality, adverse treatment effects, hypoglycaemia

## Abstract

The increased mortality reported with intensive glycaemic control has been attributed to an increased risk of treatment-related hypoglycaemia. This study investigated the relationships of haemoglobin (Hb) A_1c_, anti-hyperglycaemic treatment, and potential risks of adverse effects with all-cause mortality in patients with type 2 diabetes. Patients (*n* = 15,773) were stratified into four categories according to baseline HbA_1c_ and then assigned to three target categories, based on whether HbA_1c_ was ≤0.5% below or above (on-target), >0.5% below (below-target) or >0.5% above (above-target) their HbA_1c_ goal, personalized according to the number of potential risks among age > 70 years, diabetes duration > 10 years, advanced complication(s), and severe comorbidity (ies). The vital status was retrieved for 15,656 patients (99.26%). Over a 7.4-year follow-up, mortality risk was increased among patients in the highest HbA_1c_ category (≥8.5%) (adjusted hazard ratio, 1.34 (95% confidence interval, 1.22–1.47), *p* < 0.001) and those above-target (1.42 (1.29–1.57), *p* < 0.001). Risk was increased among individuals in the lowest HbA_1c_ category (<6.5%) and those below-target only if treated with agents causing hypoglycaemia (1.16 (1.03–1.29), *p* = 0.01 and 1.10 (1.01–1.22), *p* = 0.04, respectively). These data suggest the importance of setting both upper and lower personalized HbA_1c_ goals to avoid overtreatment in high-risk individuals with type 2 diabetes treated with agents causing hypoglycaemia.

## 1. Introduction

The impact of strict glycaemic control on excess morbidity and mortality from cardiovascular disease (CVD) in individuals with type 2 diabetes is still a matter of debate [1]. Indeed, the post-trial follow-up of the United Kingdom Prospective Diabetes Study (UKPDS) showed a significant reduction of macrovascular outcomes, together with the persistence of microvascular benefits, in patients originally randomized to intensive treatment, thus supporting the need for strict glycaemic control since the early stage of the disease [2]. Conversely, other landmark intervention trials designed to achieve more ambitious glycaemic targets, the Action to Control Cardiovascular Risk in Diabetes (ACCORD) [3], the Action in Diabetes and Vascular disease: preterAx and diamicroN-MR Controlled Evaluation (ADVANCE) [4], and the Veterans Affairs Diabetes Trial (VADT) [5], were successful in reducing the burden from microvascular disease, but failed to detect significant effects on the primary composite CVD outcome. In addition, in the intensive treatment arm, the ACCORD reported an increase in all-cause and CVD mortality, which led to the anticipated end of the trial [3], whereas the VADT showed a non-significant increment of the death rate [5]. These contrasting results have been related to differences in patients’ baseline clinical features between the UKPDS and the other three trials, i.e., newly diagnosed versus longstanding type 2 diabetes, younger versus older age, better versus worse glycaemic control, and low versus high CVD risk [6,7].

Altogether, the above findings prompted a substantial paradigm shift in glycaemic targets for patients with type 2 diabetes, moving from “one size fits all” to “personalized goals”. As a consequence, the recommended haemoglobin (Hb) A_1c_ level of <7.0% for all individuals was changed to HbA_1c_ values ranging from <6.5% to <8.5%, based on the presence and extent of several factors which may render the patient vulnerable to aggressive treatment, thus decreasing the benefits and increasing the harm from intensive glycaemic control [8]. The factors to be considered include age/life expectancy, disease duration, established complications, important comorbidities, risks associated with hypoglycaemia, individual attitude and expected treatment efforts, and resources and support systems.

However, the increased all-cause and CVD mortality reported among the ACCORD (and VADT) participants assigned to the intensive treatment group remains poorly understood. The extent (and velocity) of HbA_1c_ reduction from baseline and the rate of severe hypoglycaemia were higher in the intensive arms of the ACCORD and VADT than in other trials [6,7], thus prompting the hypothesis that the increased mortality was attributable to hypoglycaemic episodes associated with (rapid) achievement of more stringent HbA_1c_ targets in vulnerable individuals. However, although severe hypoglycaemia was associated with increased mortality, the risk of death associated with severe hypoglycaemia was relatively greater in the standard than in the intensive group of the ACCORD [9], ADVANCE [10], and VADT [11]. In addition, a post hoc analysis of the ACCORD trial showed that higher, not lower average HbA_1c_ was associated with greater risk of death, which was higher with the intensive than with the standard strategy only when average HbA_1c_ was >7.0% and when little or no decrease in HbA_1c_ followed treatment initiation [12]. These findings suggest that patients with the smaller response on glycaemic control and, hence, with persistently higher HbA_1c_ requiring more aggressive treatment, including insulin, are those at higher risk of hypoglycaemia and death, pointing to a more complex relationship among hypoglycaemia, achieved HbA_1c_, and treatment intensity.

This study aimed to investigate the relationships of on-treatment HbA_1c_ levels, type of anti-hyperglycaemic treatment, and potential risks of adverse treatment effects with all-cause mortality in individuals with type 2 diabetes. To this end, we analysed the data from participants in the Renal Insufficiency and Cardiovascular Events (RIACE) Italian Multicentre Study, who were evaluated at baseline in the years 2006–2008, when the recommended HbA_1c_ target was <7.0% for all patients.

## 2. Materials and Methods

### 2.1. Design

The RIACE is an observational, prospective, multicentre, cohort study on the impact of estimated glomerular filtration rate (eGFR) on morbidity and mortality in patients with type 2 diabetes [13]. The study was conducted in accordance with the Declaration of Helsinki. It was approved by the locally appointed ethics committees, and participants gave informed consent. Trial Registration: NCT00715481; www.ClinicalTrials.gov.

### 2.2. Subjects

The study population included 15,773 Caucasian patients (after excluding 160 individuals with missing or implausible values), consecutively attending 19 hospital-based, tertiary referral Diabetes Clinics of the National Health Service throughout Italy (see Appendix A) in the years 2006–2008. Exclusion criteria were dialysis or renal transplantation. Traditional CVD risk factors and complications were determined as part of the baseline assessment using a standardized protocol across participating centres [13].

### 2.3. All-Cause Mortality

The vital status of study subjects on 31 October 2015 was verified by interrogating the Italian Health Card database (http://sistemats1.sanita.finanze.it/wps/portal/), which provides updated and reliable information on all current Italian residents [14].

### 2.4. Traditional Cardiovascular Disease (CVD) Risk Factors

The study subjects underwent a structured interview in order to collect the following information: age at the time of the interview, smoking status, known diabetes duration, co-morbidities, and current glucose-, lipid-, and blood pressure (BP)-lowering therapy [13].

Body mass index (BMI) was calculated from weight and height. Waist circumference was measured in 4618 subjects and estimated in the remaining 11,155 individuals from the log-transformed BMI values, as previously described [15]. BP was measured with a sphygmomanometer with the patients seated with the arm at the heart level.

HbA_1c_ was measured by high-performance liquid chromatography (HPLC) using Diabetes Control and Complications Trial (DCCT)-aligned methods; triglycerides and total and high-density lipoprotein (HDL) cholesterol were determined in fasting blood samples by colorimetric enzymatic methods; non-HDL cholesterol was calculated by the formula: total cholesterol–HDL cholesterol; and low-density lipoprotein (LDL) cholesterol was calculated by the Friedewald formula. HbA_1c_ variability was calculated for each patient as the intra-individual standard deviation (HbA_1c_-SD) of 3-to-5 (4.52 ± 0.76) HbA_1c_ values obtained in 8252 individuals from 9 centres during the 2-year period preceding recruitment, including the enrolment visit [14].

### 2.5. Complications

The presence of diabetic kidney disease (DKD) was assessed by measuring albuminuria and serum creatinine, as previously detailed [13,16]. Albumin excretion rate was obtained from 24-h urine collections or calculated from the albumin-to-creatinine ratio in early-morning, first-voided urine samples, using a conversion formula developed in patients with type 1 diabetes and preliminarily validated in a subgroup of RIACE participants. Albuminuria was measured in fresh urine samples by immunonephelometry or immunoturbidimetry, in the absence of interfering clinical conditions. One-to-three measurements for each patient were obtained; in cases of multiple measurements, the geometric mean of 2–3 values was used for analysis. In individuals with multiple measurements, the concordance rate between the first value and the geometric mean was >90% for all albuminuria categories [16]. Serum (and urine) creatinine was measured by the modified Jaffe method and eGFR was calculated by the Chronic Kidney Disease Epidemiology Collaboration equation [13]. Patients were then classified into Kidney Disease: Improving Global Outcomes categories of albuminuria (A1 to A3) and eGFR (G1 to G5) and assigned to one of the following DKD phenotypes: no DKD (i.e., A1G1-A1G2), albuminuria alone (albuminuric DKD with preserved eGFR, i.e., A2G1-A2G2-A3G1-A3G2), reduced eGFR alone (non-albuminuric DKD, i.e., A1G3-A1G4-A1G5), or both albuminuria and reduced eGFR (albuminuric DKD with reduced eGFR, i.e., A2G3-A2G4-A2G5-A3G3-A3G4-A3G5), as previously reported [13].

In each centre, the presence of diabetic retinopathy (DR) was assessed by an expert ophthalmologist by dilated fundoscopy. Patients with mild or moderate non-proliferative DR were classified as having non-advanced DR, whereas those with severe non-proliferative DR, proliferative DR, or maculopathy were grouped into the advanced DR category. DR grade was assigned based on the worse eye [17].

Previous major acute CVD events, including myocardial infarction; stroke; foot ulcer/gangrene/amputation; and coronary, carotid, and lower limb revascularization, were adjudicated based on hospital discharge records by an ad hoc committee in each centre [18].

### 2.6. Categorization of Patients

Patients were stratified into the following HbA_1c_ categories according to their baseline HbA_1c_ value: <6.5% (C1); 6.5–7.49% (C2); 7.5–8.49% (C3); and ≥8.5% (C4).

In addition, patients were arbitrarily assigned the following personalized HbA_1c_ goals: <6.5% (G0); <7.0% (G1); <7.5% (G2); <8.0% (G3); and <8.5% (G4). HbA_1c_ goal were personalized according to the number of potential risks of treatment adverse effects among age > 70 years; known diabetes duration > 10 years; presence of advanced complication(s), i.e., advanced DKD (eGFR < 30 mL/min/1.73 m^2^ and/or macroalbuminuria) and/or advanced DR (severe non-proliferative, proliferative, or maculopathy) and/or history of major acute CVD event(s) (myocardial infarction, stroke, foot ulcer/gangrene/amputation, and coronary, carotid and lower limb revascularization); the presence of severe comorbidity(ies) (chronic obstructive pulmonary disease, chronic liver disease and/or cancer). Specifically, the HbA_1c_ goal was set at <6.5% if none of above risks was present (G0) and was increased by 0.5% for each of these four risks (i.e., from <7.0% (G1), if only one was present, to <8.5% (G4), if all four were present).

Patients were then assigned to three target categories, based on whether their baseline HbA_1c_ value was ≤0.5% below or above (on-target, T1), >0.5% below (below-target, T2), or >0.5% above (above-target, T3) their personalized HbA_1c_ goal.

Finally, patients from each HbA_1c_ category (C1–C4) or target category (T1–T3) were further classified based on whether they (a) were treated with anti-hyperglycaemic drugs causing hypoglycaemia (*n* = 9830), i.e., insulin and/or insulin secretagogues (sulfonylureas or glinides), either alone or combined with other anti-hyperglycaemic drugs, or (b) were not treated with these agents (*n* = 5826) and were either on lifestyle measures only (*n* = 2123, 36.4%), i.e., diet and physical activity, or receiving drugs not causing hypoglycaemia (*n* = 3703, 63.6%), i.e., acarbose, pioglitazone and/or metformin, the latter taken by the vast majority of these individual (97.3%).

The distribution of patients according to personalized HbA_1c_ goals and target categories is reported in Appendix A.

### 2.7. Statistical Analysis

Data are expressed as mean ± standard deviation (SD) or median (interquartile range), for continuous variables, and number of cases (percentage), for categorical variables. Comparisons among groups were performed by one-way analysis of variance (ANOVA) or Kruskal–Wallis test, according to the parametric or non-parametric distribution of continuous variables, followed by Bonferroni correction or Mann–Whitney test, respectively, for post-hoc comparisons, and by Pearson’s χ^2^ test for categorical variables.

Kaplan–Meier survival curves for all-cause mortality were calculated according to both HbA_1c_ categories (C1–C4) and target categories (T1–T3). Differences in survival rates were analysed using the log-rank statistic. Survival analyses were performed by Cox proportional hazards regression according to HbA_1c_ categories and target categories using C2 and T1, respectively, as reference category. Analyses by HbA_1c_ categories were adjusted for baseline age and gender (model 1), age and gender plus CVD risk factors, i.e., smoking habits, diabetes duration, BMI, triglycerides, total and HDL cholesterol, lipid-lowering treatment, systolic and diastolic BP, and anti-hypertensive treatment (model 2), and age, gender, and CVD risk factors plus complications, i.e., DKD phenotypes, DR grade, history of major acute CVD events, and comorbidities (model 3). Analyses by HbA_1c_ target categories were adjusted for gender and the CVD risk factors smoking habits, BMI, triglycerides, total and HDL cholesterol, lipid-lowering treatment, systolic and diastolic BP, anti-hypertensive treatment (model 1), and gender, the above CVD risk factors, and the factors considered for patients’ stratification into target categories, i.e., age, diabetes duration, complications, and comorbidities (model 2). In the subgroup of patients with 3-to-5 HbA_1c_ values obtained during the 2-year period before enrolment, all the above models were also adjusted for HbA_1c_-SD. In addition, the above analyses were conducted separately for individuals treated or not with agents causing hypoglycaemia and further dividing patients not on these agents in those treated with lifestyle measures only and those on drugs not causing hypoglycaemia. The results of these analyses were expressed as hazard ratios (HRs) and their 95% confidence intervals (CIs).

All p values were two-sided, and a *p* < 0.05 was considered statistically significant. Statistical analyses were performed using SPSS version 13.0 (SPSS Inc., Chicago, IL, USA).

## 3. Results

Valid information on vital status was retrieved for 15,656 patients (99.26% of the original cohort). At the time of the census, 12,054 (76.99%) patients were alive, whereas 3602 (23.01%) patients were deceased; the follow-up duration was 7.42 ± 2.05 years [19].

The baseline clinical features by HbA_1c_ categories are reported in Table 1 and Appendix A. The general HbA_1c_ goal of <7.0% was met by 6287 participants (40.2%), whereas 3645 (23.3%) patients achieved the more stringent HbA_1c_ goal of <6.5%. Individuals in the lowest HbA_1c_ category were younger and more frequently males, had lower diabetes duration and prevalence of complications and a more favourable CVD risk profile, and were less frequently current smokers and on anti-hyperglycaemic, lipid-lowering, and anti-hypertensive treatment (including therapy with agents causing hypoglycaemia) than those in the other HbA_1c_ categories (Table 1). In each HbA_1c_ category, age, diabetes duration, and prevalence of complications/comorbidities were higher, whereas the lipid and BP profiles were better in patients who were on agents causing hypoglycaemia than in those who were not (Appendix A). Kaplan–Meier estimates (Appendix A), and unadjusted HRs (Figure 1A) for all-cause mortality increased with increasing HbA_1c_ category, with similar values for C1 and C2; after adjustment for age and gender and further adjustment for CVD risk factors and complications/comorbidities, the HR for C3 became progressively similar to those of C1 and C2, whereas that for C4 remained significantly higher (Figure 1B–D). The same trend was observed among patients not treated with agents causing hypoglycaemia (Appendix A and Table 2). Conversely, among those on treatment with these drugs, Kaplan-Meier estimates and unadjusted and adjusted HRs were highest in individuals in the lowest (C1) and highest (C4) HbA_1c_ categories (Appendix A and Figure 2) and the HRs remained significantly higher versus the reference category C2 after adjustment for all confounders in both C1 and C4 (1.16 (1.03–1.29), *p* = 0.01, and 1.25 (1.13–1.39), *p* < 0.001, respectively). When analysed separately, mortality risk was higher in patients on insulin (alone or in combination) than in those on insulin secretagogues in each HbA_1c_ category, whereas it was not increased in patients treated with agents not causing hypoglycaemia and falling in the lowest HbA_1c_ category, as compared with those on lifestyle measures only. Inclusion of HbA_1c_-SD as a covariate in model 3 showed that HbA_1c_ variability was independently associated with an increased mortality risk (1.30 (1.20–1.40), *p* < 0.001, and 1.26 (1.19–1.36), *p* < 0.001, without and with patients’ stratification for type of anti-hyperglycaemic treatment, respectively), but the HRs for HbA_1c_ categories and HbA_1c_ target categories were not affected.

As expected, the great majority of participants (75.4%) had at least one potential risk (age ≥ 70 years: 38.7%; diabetes duration > 10 years: 50.3%; presence of advanced complication (s): 32.0%; and presence of severe comorbidity (ies): 17.8%) and, hence, were assigned to a higher HbA_1c_ goal. The baseline clinical features by HbA_1c_ target categories are reported in Table 3 and Appendix A. Participants with above-target HbA_1c_ values were 6,046 (38.6%), whereas the remaining 9610 patients (61.4%) achieved their personalized HbA_1c_ goal; however, of them, 4989 (31.9%) were on-target, whereas 4621 (29.5%) were below-target, i.e., showed HbA_1c_ values > 0.5% lower than their personalized goal. As expected, individuals who were below-target had a higher prevalence of potential risks (especially older age and higher prevalence of complications/comorbidities) and, hence, were more frequently assigned the highest personalized HbA_1c_ goals (i.e., <7.5% or higher, G2–G4), than both on-target and above-target patients (59.6% versus 40.7% and 37.0%, respectively) (Appendix A). In addition, below-target individuals were more frequently males and less frequently on insulin and agents causing hypoglycaemia and showed a better CVD risk profile than above-target patients (Table 3). In each HbA_1c_ target category, potential risks, i.e., age, diabetes duration, and prevalence of complications/comorbidities, were higher in patients who were on agents causing hypoglycaemia than in those who were not (Appendix A). Kaplan–Meier estimates (Appendix A) and unadjusted HRs (Figure 3A) for all-cause mortality were lowest in patients on-target, intermediate in those above-target, and highest in those below-target. The increased risk in below-target individuals was progressively attenuated by adjustment for CVD risk factors and potential risks, whereas it became significantly elevated in above-target patients (Figure 3B,C). All the other covariates included in model 3 were significantly associated with mortality, except BMI and systolic BP, and sensitivity analysis showed that, among potential risks, age had the highest weight. The same trend was observed among patients not treated with agents causing hypoglycaemia (Appendix A and Table 4). Conversely, among patients on treatment with these drugs, the increased risk of death versus the reference category T1 remained after adjustment for CVD risk factors and potential risks in both T2 (1.10 (1.01–1.22), *p* = 0.04) and T3 (1.20 (1.09–1.32), *p* < 0.001) individuals. Again, when analysed separately, mortality risk was higher in individuals on insulin (alone or in combination) than in those on insulin secretagogues in each HbA_1c_ target category and was not increased in patients treated with agents not causing hypoglycaemia and falling in the below-target category, as compared with those on lifestyle measures only. Results did not change when including HbA_1c_-SD among the covariates or when HbA_1c_ goals were personalized using different cut-offs for age (>75 years) and diabetes (>15 years).

## 4. Discussion

This analysis of individuals with type 2 diabetes from the RIACE cohort provides evidence that achieving HbA_1c_ levels close to normal values (i.e., <6.5%) is not associated with increased mortality, unless patients are on treatment with drugs causing hypoglycaemia. These finding are consistent with the positive relationship between severe hypoglycaemia and increased mortality reported in the ACCORD [9], ADVANCE [10], and VADT [11] and with the observation that, in the ACCORD, risk of death was higher in the intensive than in the conventional group only in patients requiring more aggressive treatment, including insulin, because of little or no decrease in HbA_1c_ following treatment initiation [12]. However, unmeasured factors other than hypoglycaemia may be involved in the increased mortality in patients on treatment with agents causing hypoglycaemia, including poor adherence, depression, cognitive impairment, education, socio-economic status, etc. The results of our study are also consistent with the increased mortality reported in older people with type 2 diabetes from the Fremantle Diabetes Study Phase II who were treated with sulfonylurea and/or insulin and had HbA_1c_ levels < 7.0%, but not with the increased mortality observed in those who were treated with metformin and had HbA_1c_ levels < 6.5%, a finding which the authors attributed to confounding by indication [20]. Conversely, our data are in contrast with previous reports from the UK General Practice Research Database [21] and the Kaiser Permanente Diabetes Registry of Northwest [22] and Northern California [23] showing a U-shape relationship between achieved HbA_1c_ and mortality in diabetic individuals, with both higher and lower HbA_1c_ values associated with increased all-cause mortality. However, Currie et al. analysed patients whose treatment had been intensified from oral monotherapy to combination therapy with oral agents or to regimens that included insulin; in addition, those with lower mean HbA_1c_ levels were older and had worse renal function than those with higher mean HbA_1c_ values [21]. Indeed, the 2008 American Diabetes Association (ADA) guidelines recommended less stringent HbA_1c_ goals when the incremental but small benefit from lowering HbA_1c_ from 7.0% to 6.0% [24,25] may be outweighed by the increased risk of hypoglycaemia and the great effort required to achieve near-normoglycaemia [26]. Subsequently, based on the results of the ACCORD [3], ADVANCE [4], and VADT [5], the 2012 Position Statement of the ADA and the European Association for the Study of Diabetes recommended a patient-centred approach with setting of personalized HbA_1c_ goals according to several potential risks of adverse treatment effects [8].

When patients from the RIACE cohort were arbitrarily assigned to five different HbA_1c_ goals (from <6.5% to <8.5%) according the number (0 to 4) of potential risks among age > 70 years, diabetes duration >10 years, presence of advanced complication and severe comorbidities, 44.8% of patients were reclassified to a higher target, whereas 24.6% of patients were reclassified to a lower target than the general goal of <7.0%. As a consequence, the proportion of individuals with HbA_1c_ values less than the personalized goal (61.4%) was higher than that of patients with HbA_1c_ values less than the general < 7.0% goal (40.2%) and much higher than that of patients meeting the more stringent HbA_1c_ goal of <6.5% (23.3%), consistent with previous reports from the National Health and Nutrition Examination Survey (NHANES) [27,28]. In addition, about half of the RIACE participants with HbA_1c_ levels less than their personalized goal (29.5% of the whole cohort) were well below this threshold (i.e., >0.5%), as the personalized goal for the majority of them was higher than the general < 7.0% goal which was recommended at the time patients underwent baseline evaluation (i.e., in the years 2006–2008). These findings are consistent with a cross-sectional analysis of the data from older (≥65 years) NHANES participants with diabetes from 2001 through 2010, which showed that a substantial proportion of patients were potentially overtreated, i.e., had an HbA_1c_ < 7.0% irrespective of health status and type of treatment, with no change over the 10-year study period [29]. In addition, a cross-sectional analysis of 42,669 outpatients with type 2 diabetes from the Diabetes Collaborative Registry showed that one-fourth were tightly controlled with medications that confer a high risk of hypoglycaemia, thus suggesting potential overtreatment of a substantial proportion of people [30]. Finally, a retrospective cohort study using data from the US Veterans Health Administration showed that only one-fourth of the older patients whose treatment resulted in very low levels of HbA_1c_ or BP underwent de-intensification of anti-hyperglycaemic or anti-hypertensive therapy [31].

The most intriguing and original finding of our study is that below-target patients showed a mortality risk higher than that of on-target individuals (i.e., with HbA_1c_ values ≤ 0.5% below or above their personalized goal) and similar to that of individuals above-target (i.e., with HbA_1c_ values > 0.5% above their personalized goal). Excess risk was observed in the whole cohort and in patients on treatment with anti-hyperglycaemic drugs causing hypoglycaemia, but not in patients not treated with these agents, and was attenuated, though not completely in those on “hypoglycaemic” agents, after adjusting for CVD risk factors and the potential risks considered for patients’ stratification into target categories.

Our findings that, in patients on treatment with agents causing hypoglycaemia, HbA_1c_ values < 6.5% or ≥0.5% less than the personalized goal are associated with increased mortality risk have important implications for clinical practice. First, these data support the importance of setting a personalized HbA_1c_ range rather than an upper HbA_1c_ threshold (e.g., 7.0–7.5% instead of <7.5%) when the use of agents causing hypoglycaemia is required for achieving the glycaemic goal. This approach would be in line with that adopted by the recent guidelines of the European Society of Cardiology (in collaboration with the European Association for the Study of Diabetes (EASD)) for the treatment of hypertension in diabetic patients, which recommend a systolic BP goal of <130 mmHg, but not <120 mmHg, in younger individuals and of <140 mmHg, but not <130 mmHg, in older individuals and a diastolic BP goal of <80 mmHg, but not <70 mmHg in all patients [32]. Second, our results provide further support to the current guideline recommendations to set higher glycaemic goals when the risks of lower HbA_1c_ targets may outweigh the potential benefits and to modify the treatment regimen accordingly [33]. Treatment modification may include de-intensification of pharmacologic therapy with drugs causing hypoglycaemia in patients with HbA_1c_ levels well below the individual HbA_1c_ goal. At the time patients underwent baseline evaluation, only a few therapeutic options (i.e., metformin, pioglitazone, and acarbose) were available in addition to insulin and insulin secretagogues which, therefore, were used by a substantial proportion of the RIACE participants (62.8%). However, during the last few decades, several new classes of anti-hyperglycaemic agents that do not cause hypoglycaemia have been made available, possibly allowing more stringent HbA_1c_ goals even in high-risk patients, provided that combination therapy with insulin and/or insulin secretagogues is not required for achieving these targets.

The strengths of this study include the large size of the cohort, the analysis of a contemporary real-life dataset, the assessment of a wide range of clinical parameters, and the completeness of baseline and follow-up data. However, this study has several limitations. First, the analysis is based on baseline HbA_1c_ levels and treatments, which have likely changed during the follow-up for several reasons, including disease progression, guideline change, and availability of new drugs, the use of which was however very limited at the time of the census. Second, no data are available on the number and severity of hypoglycaemic episodes as well as on individual attitude and expected treatment efforts and resources and support systems. Third, the score (0 to 4) used for personalized HbA_1c_ goals may not mirror the severity of complications and comorbidities, as it does not distinguish one complication or comorbidity from another and individuals with one or multiple complications or comorbidities. Fourth, the study findings may not be applicable to the general ambulatory diabetes population, as only part of the individuals with type 2 diabetes attend tertiary referral outpatients Diabetes Clinics in Italy. Fifth, the observational design makes causal interpretation impossible. Sixth, there are potential methodological limitations, which have been extensively addressed in previous RIACE reports [13,14,15,16,17,18,19].

## 5. Conclusions

This analysis of the RIACE cohort indicates that near-normal HbA_1c_ levels or HbA_1c_ values well below the personalized goal are associated with an increased risk of death only if achieved with the use of agents causing hypoglycaemia. These findings support the importance of avoiding overtreatment with these drugs in high-risk individuals by setting both upper and lower personalized HbA_1c_ goals and of using anti-hyperglycaemic agents that do not cause hypoglycaemia for safely achieving more stringent HbA_1c_ goals.

## Figures and Tables

**Figure 1 jcm-09-00246-f001:**
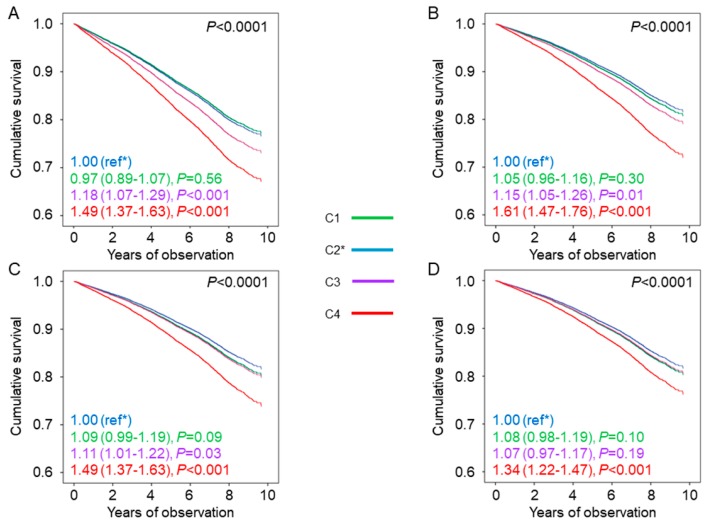
Mortality risk by HbA_1c_ categories. Cox proportional hazards regression according to HbA_1c_ categories, unadjusted (**A**) and adjusted for age and gender (**B**), plus cardiovascular disease (CVD) risk factors (**C**), plus complications/comorbidities (**D**). HRs (95% CI) for mortality are shown for each HbA_1c_ category. HbA_1c_ = haemoglobin A_1c_; HR = hazard ratio; CI = confidence interval; C1 = HbA_1c_ <6.5%; C2 = HbA_1c_ 6.5%–7.49%; C3 = HbA_1c_ 7.5%–8.49%; C4 = HbA_1c_ ≥8.5%; ref*, the asterisk indicates the reference category.

**Figure 2 jcm-09-00246-f002:**
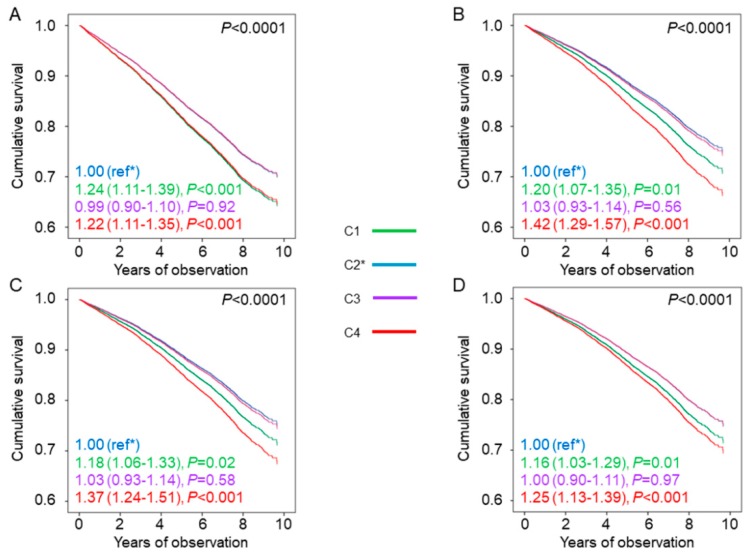
Mortality risk by HbA_1c_ categories and use of agents causing hypoglycaemia. Cox proportional hazards regression according to HbA_1c_ categories among participants treated with agents causing hypoglycaemia, unadjusted (**A**) and adjusted for age and gender (**B**), plus CVD risk factors (**C**), plus complications/comorbidities (**D**). HRs (95% CI) for mortality are shown for each HbA_1c_ category. HbA_1c_ = haemoglobin A_1c_; HR = hazard ratio; CI = confidence interval; C1 = HbA_1c_ < 6.5%; C2 = HbA_1c_ 6.5–7.49%; C3 = HbA_1c_ 7.5–8.49%; C4 = HbA_1c_ ≥ 8.5%.

**Figure 3 jcm-09-00246-f003:**
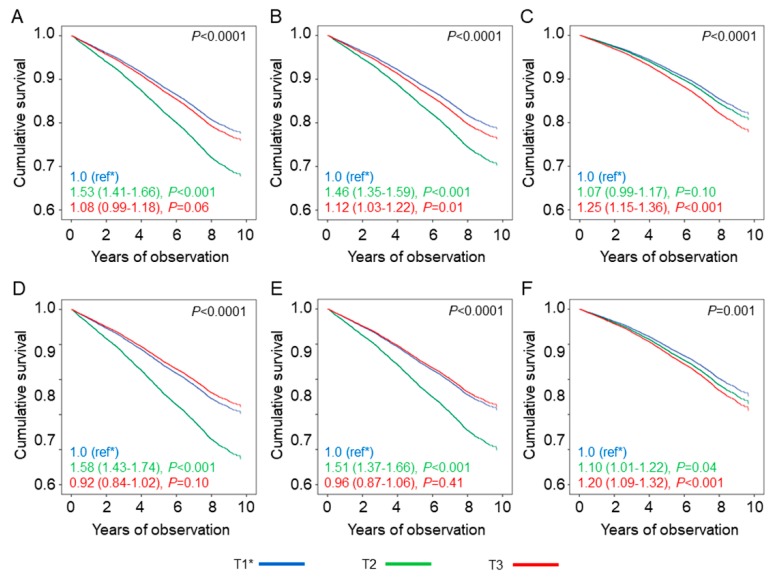
Mortality risk by HbA_1c_ target categories and use of agents causing hypoglycaemia. Cox proportional hazards regression according to HbA_1c_ target categories in the whole cohort (**A**–**C**) and among participants treated with agents causing hypoglycaemia (**D**–**F**), unadjusted (**A**,**D**) and adjusted for CVD risk factors (**B**,**E**), complications/comorbidities (**C**,**F**). HRs (95% CI) for mortality are shown for each HbA_1c_ target category. HbA_1c_ = haemoglobin A_1c_; HR = hazard ratio; CI = confidence interval; T1 = HbA_1c_ on-target (≤0.5% below or above personalized goal); T2 = HbA_1c_ below-target (>0.5% below personalized goal); T3 = HbA_1c_ above-target (>0.5% above personalized goal).

**Table 1 jcm-09-00246-t001:** Baseline clinical features of study participants by HbA_1c_ categories.

Variables	C1	C2	C3	C4	*p*-Value
N (%)	3645 (23.3)	5081 (32.5)	3608 (23.0)	3322 (21.2)	
Deaths, *n* (%)	737 (20.2)	1049 (20.6)	858 (23.8)	958 (28.8)	<0.001
Age, years	65.6 ± 10.6	67.1 ± 9.9	67.1 ± 10.2	66.3 ± 10.8	<0.001
Male gender, *n* (%)	2233 (57.1)	2903 (55.9)	2018 (52.6)	1748 (56.9)	<0.001
Smoking, *n* (%)					<0.001
Never	2021 (55.4)	2850 (56.1)	2090 (57.9)	1888 (56.8)	
Former	1107 (30.4)	1472 (29.0)	960 (26.6)	868 (26.1)	
Current	517 (14.2)	759 (14.9)	558 (15.5)	566 (17.0)	
Diabetes duration, years	9.0 ± 8.9	12.5 ± 9.9	15.7 ± 10.1	16.1 ± 10.2	<0.001
HbA_1c_, %	5.93 ± 0.46	6.97 ± 0.29	7.93 ± 0.28	9.77 ± 1.31	<0.001
(mmol·mol^−1^)	(41.3 ± 5.0)	(52.7 ± 3.2)	(63.2 ± 3.1)	(83.3 ± 14.3)	
BMI, kg·m^−2^	28.6 ± 4.9	28.6 ± 4.9	29.1 ± 5.2	29.7 ± 5.5	<0.001
Waist circumference, cm	101.8 ± 10.0	101.9 ± 10.0	102.7 ± 10.6	104.0 ± 11.1	<0.001
Triglycerides, mmol·L^−1^	1.46 ± 0.94	1.49 ± 0.85	1.56 ± 0.89	1.84 ± 1.27	<0.001
Total cholesterol, mmol L^−1^	4.74 ± 0.96	4.77 ± 0.95	4.75 ± 0.96	4.89 ± 1.09	<0.001
HDL cholesterol, mmol·L^−1^	1.31 ± 0.36	1.31 ± 0.35	1.28 ± 0.34	1.24 ± 0.35	<0.001
LDL cholesterol, mmol L^−1^	3.43 ± 0.92	3.46 ± 0.91	3.47 ± 0.92	3.65 ± 1.05	0.01
Non-HDL cholesterol, mmol L^−1^	2.78 ± 0.84	2.79 ± 0.83	2.76 ± 0.82	2.83 ± 0.90	<0.001
Systolic BP, mmHg	136.3 ± 17.5	137.9 ± 17.7	139.6 ± 18.2	138.6 ± 18.7	<0.001
Diastolic BP, mmHg	79.0 ± 9.4	78.6 ± 9.2	78.9 ± 9.5	78.7 ± 9.7	0.16
Pulse pressure, mmHg	57.3 ± 15.3	59.3 ± 15.6	60.8 ± 16.0	60.0 ± 15.8	<0.001
Anti-hyperglycaemic treatment, *n* (%)					
Lifestyle	1017 (27.9)	762 (15.0)	192 (5.3)	142 (4.3)	<0.001
Insulin	440 (12.1)	913 (18.0)	1062 (29.4)	1509 (45.4)	<0.001
Non-insulin agents	2188 (60.0)	3406 (67.0)	2354 (65.2)	1671 (50.3)	<0.001
Metformin	1746 (47.9)	2840 (55.9)	2206 (61.1)	1853 (55.8)	<0.001
Pioglitazone	87 (2.4)	167 (3.3)	163 (4.5)	137 (4.1)	<0.001
Acarbose	36 (1.0)	43 (0.8)	47 (1.3)	44 (1.3)	0.09
Sulfonylureas	820 (22.5)	1677 (33.0)	1472 (40.8)	1281 (38.6)	<0.001
Repaglinide	335 (9.2)	537 (10.6)	371 (10.3)	282 (8.5)	0.01
Agents causing hypoglycaemia, *n* (%)	1538 (42.2)	2944 (57.9)	2646 (73.3)	2702 (81.3)	<0.001
Lipid-lowering treatment, *n* (%)	1572 (43.1)	2389 (47.0)	1728 (47.9)	1549 (46.6)	<0.001
Anti-hypertensive treatment, *n* (%)	2509 (68.8)	3611 (71.1)	2601 (72.1)	2351 (70.8)	<0.001
Albuminuria, mg·day^−1^	53.8 ± 225.5	68.7 ± 373.2	74.4 ± 329.7	96.0 ± 293.1	<0.001
Serum creatinine, μmol·L^−1^	81.3 ± 38.0	80.4 ± 35.4	80.4 ± 31.8	82.2 ± 31.8	0.27
eGFR, mL·min^−1^·1.73 m^−2^	81.6 ± 21.0	80.3 ± 20.1	79.9 ± 20.5	79.2 ± 22.6	<0.001
DKD phenotype, *n* (%)					<0.001
No DKD	2554 (70.1)	3385 (66.6)	2269 (62.9)	1776 (53.5)	
Albuminuric DKD with preserved eGFR	532 (14.6)	877 (17.3)	691 (19.2)	866 (26.1)	
Non-albuminuric DKD	321 (8.8)	465 (9.2)	360 (10.0)	330 (9.9)	
Albuminuric DKD with reduced eGFR	238 (6.5)	354 (7.0)	288 (8.0)	350 (10.5)	
DR, *n* (%)					<0.001
No DR	3178 (87.2)	4185 (82.4)	2658 (73.7)	2168 (63.5)	
Non-advanced DR	242 (6.6)	503 (9.9)	586 (16.2)	616 (18.5)	
Advanced DR	225 (6.2)	393 (7.7)	364 (10.1)	538 (16.2)	
CVD, *n* (%)					
Any	699 (19.2)	1086 (21.4)	906 (25.1)	929 (28.0)	<0.001
Myocardial infarction	354 (9.7)	514 (10.1)	445 (12.3)	429 (12.9)	<0.001
Coronary revascularization	295 (8.1)	487 (9.6)	411 (11.4)	386 (11.6)	<0.001
Stroke	126 (3.5)	143 (2.8)	118 (3.3)	126 (3.8)	0.09
Carotid revascularization	120 (3.3)	240 (4.7)	235 (6.5)	261 (7.9)	<0.001
Ulcer/gangrene/amputation	92 (2.5)	167 (3.3)	131 (3.6)	166 (5.0)	<0.001
Lower limb revascularization	65 (1.8)	135 (2.7)	121 (3.4)	129 (3.9)	<0.001
Comorbidities, *n* (%)					
Any	682 (18.7)	872 (17.2)	620 (17.2)	613 (18.5)	0.15
COPD	173 (4.7)	201 (4.0)	142 (3.9)	158 (4.8)	<0.001
Chronic liver disease	299 (8.2)	415 (8.2)	326 (9.0)	321 (9.7)	0.06
Cancer	279 (7.7)	342 (6.7)	215 (6.0)	195 (5.9)	<0.001

HbA_1c_ = haemoglobin A_1c_; C1 = HbA_1c_ <6.5%; C2 = HbA_1c_ 6.5–7.49%; C3 = HbA_1c_ 7.5–8.49%; C4 = HbA_1c_ ≥8.5%; BMI = body mass index; HDL = high-density lipoprotein; LDL = low-density lipoprotein; BP = blood pressure; ACE-I = angiotensin converting enzyme inhibitor; ARB = angiotensin receptor blocker; eGFR = estimated glomerular filtration rate; DKD = diabetic kidney disease; DR = diabetic retinopathy; CVD = cardiovascular disease; COPD = chronic obstructive pulmonary disease.

**Table 2 jcm-09-00246-t002:** Mortality risk by HbA_1c_ categories among participants not treated with agents causing hypoglycaemia.

HbA_1c_ Target Categories	Unadjusted			Adjusted								
				Model 1			Model 2			Model 3		
	HR	95% CI	*p*-Value	HR	95% CI	*p*-Value	HR	95% CI	*p*-Value	HR	95% CI	*p*-Value
**C2 (ref)**	1.00	-	<0.001	1.00	-	<0.001	1.00	-	<0.001	1.00	-	<0.001
**C1**	0.92	0.78–1.10	0.36	1.01	0.85–1.20	0.94	1.03	0.87–1.23	0.70	1.03	0.87–1.22	0.74
**C3**	1.36	1.12–1.66	0.01	1.34	1.10–1.63	0.01	1.28	1.05–1.55	0.02	1.26	1.04–1.54	0.02
**C4**	1.67	1.35–2.07	<0.001	1.72	1.39–2.13	<0.001	1.59	1.28–1.97	<0.001	1.50	1.21–1.86	<0.001

Cox proportional hazards regression according to HbA_1c_ categories plus use of agents causing hypoglycaemia, unadjusted and adjusted for age and gender (model 1), plus CVD risk factors (model 2), plus complications/comorbidities (model 3). HRs (95% CI) for mortality are shown for each HbA_1c_ category. HbA_1c_ = haemoglobin A_1c_; HR = hazard ratio; CI = confidence interval; C1 = HbA_1c_ < 6.5%; C2 = HbA_1c_ 6.5–7.49%; C3 = HbA_1c_ 7.5–8.49%; C4 = HbA_1c_ ≥ 8.5%.

**Table 3 jcm-09-00246-t003:** Baseline clinical features of study participants by HbA_1c_ target categories.

Variables	T1	T2	T3	*p*-Value
*n* (%)	4989 (31.9)	4621 (29.5)	6046 (38.6)	
Deaths, *n* (%)	991 (19.9)	1330 (28.8)	1281 (21.2)	<0.001
Age, years	66.7 ± 9.9	69.4 ± 10.1	64.4 ± 10.3	<0.001
Male gender, *n* (%)	2833 (56.8)	2818 (61.0)	3251 (53.8)	<0.001
Smoking, *n* (%)				<0.001
Never	2837 (56.9)	2581 (55.9)	3431 (56.7)	
Former	1435 (28.8)	1455 (31.5)	1517 (25.1)	
Current	717 (14.4)	585 (12.7)	1098 (18.2)	
Diabetes duration, years	12.7 ± 10.1	13.3 ± 10.7	13.6 ± 9.8	<0.001
HbA_1c_, %	7.13 ± 0.57	6.22 ± 0.66	8.90 ± 1.40	<0.001
(mmol·mol^−1^)	(54.4 ± 6.2)	(44.5 ± 7.2)	(73.8 ± 15.3)	
BMI, kg·m^−2^	28.8 ± 5.0	28.2 ± 4.7	29.7 ± 5.5	<0.001
Waist circumference, cm	102.1 ± 10.1	101.1 ± 9.7	103.9 ± 11.0	<0.001
Triglycerides, mmol·L^−1^	1.51 ± 0.89	1.42 ± 0.86	1.74 ± 1.14	<0.001
Total cholesterol, mmol·L^−1^	4.78 ± 0.95	4.69 ± 0.96	4.86 ± 1.04	<0.001
HDL cholesterol, mmol·L^−1^	1.30 ± 0.34	1.32 ± 0.37	1.26 ± 0.35	<0.001
LDL cholesterol, mmol·L^−1^	3.47 ± 0.91	3.37 ± 0.91	3.61 ± 1.00	<0.001
Non-HDL cholesterol, mmol·L^−1^	2.79 ± 0.82	2.73 ± 0.83	2.83 ± 0.88	<0.001
Systolic BP, mmHg	138.3 ± 17.9	137.5 ± 17.9	138.2 ± 18.2	0.06
Diastolic BP, mmHg	79.0 ± 9.2	78.3 ± 9.5	78.9 ± 9.6	0.01
Pulse pressure, mmHg	59.4 ± 15.6	59.2 ± 15.8	59.3 ± 15.7	0.87
Anti-hyperglycaemic treatment, *n* (%)				
Lifestyle	810 (16.2)	927 (20.1)	376 (6.2)	<0.001
Insulin	972 (19.5)	860 (18.6)	2092 (34.6)	<0.001
Non-insulin agents	3207 (64.3)	2834 (61.3)	3578 (59.2)	<0.001
Metformin	2784 (55.8)	2195 (47.5)	3666 (60.6)	<0.001
Pioglitazone	181 (3.6)	92 (2.0)	281 (4.6)	<0.001
Acarbose	49 (1.0)	47 (1.0)	74 (1.2)	0.41
Sulfonylureas	1610 (32.3)	1314 (28.4)	2326 (38.5)	<0.001
Repaglinide	461 (9.2)	510 (11.0)	554 (9.2)	0.01
Agents causing hypoglycaemia, *n* (%)	2835 (56.8)	2557 (55.3)	4438 (73.4)	<0.001
Lipid-lowering treatment, *n* (%)	2314 (46.4)	2148 (46.5)	2776 (45.9)	0.82
Anti-hypertensive treatment, *n* (%)	3504 (70.2)	3449 (74.6)	4119 (68.1)	<0.001
Albuminuria, mg·day^−1^	64.1 ± 285.4	80.3 ± 377.0	73.0 ± 290.3	0.04
Serum creatinine, μmol·L^−1^	79.6 ± 31.8	85.7±42.4	78.7±29.2	<0.001
eGFR, mL·min^−1^·1.73 m^−2^	81.0±19.7	76.6±21.7	82.5±21.0	<0.001
DKD phenotype, *n* (%)				<0.001
No DKD	3348 (67.1)	2859 (61.9)	3777 (62.5)	
Albuminuric DKD with preserved eGFR	863 (17.3)	767 (16.6)	1336 (22.1)	
Non-albuminuric DKD	461 (9.2)	523 (11.3)	492 (8.1)	
Albuminuric DKD with reduced eGFR	317 (6.4)	472 (10.2)	441 (7.3)	
DR, *n* (%)				<0.001
No DR	4064 (81.5)	3695 (80.0)	4430 (73.3)	
Non-advanced DR	529 (10.6)	444 (9.6)	974 (16.1)	
Advanced DR	396 (7.9)	482 (10.4)	642 (10.6)	
CVD, *n* (%)				
Any	1004 (20.1)	1457 (31.5)	1159 (19.2)	<0.001
Myocardial infarction	480 (9.6)	703 (15.2)	559 (9.2)	<0.001
Coronary revascularization	447 (9.0	617 (13.4)	515 (8.5)	<0.001
Stroke	137 (2.7)	231 (5.0)	145 (2.4)	<0.001
Carotid revascularization	250 (5.0)	308 (6.7)	298 (4.9)	<0.001
Ulcer/gangrene/amputation	158 (3.2)	217 (4.7)	181 (3.0)	<0.001
Lower limb revascularization	122 (2.4)	171 (3.7)	157 (2.6)	<0.001
Comorbidities *n* (%)				
Any	756 (15.2)	1261 (27.3)	770 (12.7)	<0.001
COPD	167 (3.3)	322 (7.0)	185 (3.1)	<0.001
Chronic liver disease	385 (7.7)	570 (12.3)	406 (6.7)	<0.001
Cancer	276 (5.5)	504 (10.9)	251 (4.2)	<0.001

HbA_1c_ = haemoglobin A_1c_; T1 = HbA_1c_ on-target (≤0.5% below or above personalized goal); T2 = HbA_1c_ below-target (>0.5% below personalized goal); T3 = HbA_1c_ above-target (>0.5% above personalized goal); BMI = body mass index; HDL = high-density lipoprotein; LDL = low-density lipoprotein; BP = blood pressure; ACE-I = angiotensin converting enzyme inhibitor; ARB = angiotensin receptor blocker; eGFR = estimated glomerular filtration rate; DKD = diabetic kidney disease; DR = diabetic retinopathy; CVD = cardiovascular disease; COPD = chronic obstructive pulmonary disease.

**Table 4 jcm-09-00246-t004:** Mortality risk by HbA_1c_ target categories among participants not treated with agents causing hypoglycaemia.

HbA_1c_ Target Categories	Unadjusted			Adjusted					
				Model 1			Model 2		
	HR	95% CI	*p*-Value	HR	95% CI	*p*-Value	HR	95% CI	*p*-Value
T1 (ref)	1.00	-	<0.001	1.00	-	<0.001	1.00	-	<0.001
T2	1.54	1.31–1.82	<0.001	1.49	1.26–1.75	<0.001	1.08	0.92–1.28	0.34
T3	1.11	0.92–1.33	0.290	1.14	0.94–1.37	0.19	1.21	1.01–1.46	0.04

Cox proportional hazards regression according to HbA_1c_ target categories plus use of agents causing hypoglycaemia, unadjusted and adjusted for CVD risk factors (model 1) and complications/comorbidities (model 2). HRs (95% CI) for mortality are shown for each HbA_1c_ target category. HbA_1c_ = haemoglobin A_1c_; HR = hazard ratio; CI = confidence interval; T1 = HbA_1c_ on-target (≤0.5% below or above personalized goal); T2 = HbA_1c_ below-target (>0.5% below personalized goal); T3 = HbA_1c_ above-target (>0.5% above personalized goal).

## Data Availability

Data are available upon request from the corresponding author.

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
