# Peer review of "Association between On-Treatment Haemoglobin A1c and All-Cause Mortality in Individuals with Type 2 Diabetes: Importance of Personalized Goals and Type of Anti-Hyperglycaemic Treatment"

_jcm, 2020, doi:10.3390/jcm9010246_

Round 1

Reviewer 1 Report

The reviewer appreciates the authors’ response. The reviewer now understands that the authors intend to focus on the association rather than the intervention effect. Thank you. There is one comment.

#1 Confusion of association with intervention effect

The reviewer is afraid that the following description in the abstract would be misleading; “These data suggest setting both upper and lower personalized HbA1c goals and de-intensifying therapy if HbA1c falls well below target in high-risk diabetic individuals treated with agents causing hypoglycemia.” It was true that low HbA1c levels were associated with a poorer prognosis under treatment with agents causing hypoglycemia. However, it does not mean that it would be better to de-intensify therapy in patients with low HbA1c levels under treatment with agents causing hypoglycemia.
Take “cholesterol paradox”, for example. Epidemiological studies demonstrate that lower cholesterol levels were associated with an increased mortality risk. But it does not mean that intensifying anti-hyperlipidemic therapy would be harmful.

Similarly, it would be better to soften the description in the forth paragraph in the discussion section (“these data support the importance of setting a personalized HbA1c range rather than an upper HbA1c threshold (e.g., 7.0%-7.5% instead of <7.5%) when the use of agents causing hypoglycaemia is required for achieving the glycaemic goal and suggest that pharmacologic therapy with these drugs should be de-intensified if HbA1c levels are well below the individual HbA1c goal.”), as well as the description in the conclusions section (“These findings suggest to avoid overtreatment with these drugs in frail high-risk individuals by setting both upper and lower personalized HbA1c goals and  support the use of anti-hyperglycaemic agents that do not cause hypoglycaemia for safely achieving more stringent HbA1c goals.”).

Reviewer 2 Report

The paper is improved and reads much better. Now that they are providing data on non-hypoglycaemic treatment, I think they should clarify how these cases were defined. Are they including a range of antidiabetic but non-hypoglycaemic drugs? What about patients on diet alone, are they included? The paper I quoted in my original critique was specifically about metformin. Should they not cite that paper given the relevance?

Author Response

This manuscript is a resubmission of an earlier submission. The following is a list of the peer review reports and author responses from that submission.

Round 1

Reviewer 1 Report

The paper presents the associations between HbA1c and all cause mortality. The topic is interesting, and rather ambitious. 

There are several weak points:

the clustering in G0-G4 is not clear; applying an artificial technique of clustering could assure a more rigorous method of defying the groups. a combination of frailty factors should also be considered.

Reviewer 2 Report

The authors reported the association of hemoglobin A1c (HbA1c) levels with all-cause mortality in individuals with type 2 diabetes. The topic is of clinical interest, and they analyzed a database of a large prospective study. However, their statistical approaches and interpretations require extensive revisions.

#1 Confusion of association with intervention effect

As the authors declared in the manuscript, the purpose of this study was to investigate “the relationship of on-treatment HbA1c levels, type of anti-hyperglycemic treatment, and frailty factors with all-cause mortality in individuals with type 2 diabetes.” In this sense, their statistical approach, i.e., a multivariate regression analysis, was correct. However, they interpreted the revealed association as a treatment effect, which was not appropriate.

The association and the treatment effect should be carefully distinguished from each other. The authors performed a multivariate regression analysis and observed an association of HbA1c levels with all-cause mortality. However, this association did not mean that controlling HbA1c levels (i.e., intervention) would increase or decrease the future risk of mortality.

If they really believed so (i.e., an association meant an intervention effect), then how could they explain another striking result of theirs, i.e., the finding that treatment with SU, glinide, and/or insulin was associated with about 1.5-time higher risk of mortality than that without such medications, regardless of HbA1c goals, even after full adjustment? Would they argue that SU, glinide, and insulin are harmful medications regardless of HbA1c goals?

#2 Very high mortality risk of SU/glinide/insulin

They revealed that treatment with SU, glinide, and/or insulin was associated with about 1.5-time higher risk of mortality than that without such medications, regardless of HbA1c goals, even after full adjustment. A 1.5 times were very high. Based on the literature, the reviewer (and perhaps the readers of the Journal) recognizes (or believes) that SU, glinide, and insulin would not be so harmful. The authors should clearly explain this striking finding of theirs. The reviewer would like to recommend the authors not to interpret this association simply as an intervention effect. Given that the high mortality risk of SU, glinide, and insulin was not sufficiently attenuated even after full adjustment, there would be a number of unmeasured important confounding factors. In other words, the association of HbA1c levels under certain medications on the mortality risk, which the authors more strongly emphasized, was also overestimated due to insufficient adjustment for confounding factors.

#3 Analysis of intervention effect in observational study

The reviewer understands that the current study was an observational one, and not an interventional trial. Intervention effects will be ultimately proved through clinical trials, while observational studies just imply potential intervention effects. It is true that some statistical approaches, e.g., the propensity score-matching analysis, have been suggested for assessing intervention effects in an observational study. However, the current theme seemed too complex to adopt such approaches, since HbA1c levels were not only an intention to treat but also a consequence of treatment. To evaluate the intervention effect of HbA1c levels under specific medications, at least the information on pre-treatment HbA1c levels and other factors influencing the selection of medications would be required.

#4 Difference in prognostic impact between treatments

The authors found that a certain glycemic control had a significant association with the mortality risk under a specific medication regimen, whereas it did not under another regimen. Based on this finding, the authors argued that the prognostic impact was different between the two regimens. However, this was not statistically correct. If one focuses on the difference in the prognostic impact, one should evaluate the interaction effect. If the interaction effect is statistically significant, one could safely conclude that there is a difference. Statistical significance in a subgroup but not in another subgroup might just come from type I error during multiple testing.

#5 Factors to be considered in setting individual HbA1c target

As the authors mentioned in the Introduction section, the factors to be considered in setting individual HbA1c target include age/life expectancy, disease duration, established complications, important comorbidities, risks associated with hypoglycemia, individual attitude and expected treatment efforts, and resources and support systems. However, in the current study, some of them (i.e., life expectancy, risks associated with hypoglycemia, individual attitude and expected treatment efforts, and resources and support systems) were ignored. Nonetheless, the expressions in the manuscript seemed as if they considered them all. To avoid misunderstanding, it would be better to soften their tone.

#6 Estimation of albuminuria

Albumin excretion rate in some individuals was calculated from albumin-to-creatinine ratio in early-morning, first-voided urine samples, using a conversion formula developed in patients with type 1 diabetes and preliminarily validated in a subgroup of RIACE participants. Please give a detailed explanation about the conversion formula. Citing references and demonstrating the validation would be also important, because their grading of “frailty”, a key point in this study, was dependent on the judgement of albuminuria.

Furthermore, the authors mentioned that “one-to-three measurements for each patient were obtained”. When, and at what interval, were these measurements obtained? Before enrollment? Or During the 10-year follow-up period?

#7 Advanced complications

The authors determined advanced complications as advanced DKD, advanced DR, and/or history of major acute CVD events. What did “acute” CVD events mean? “Acute” CVD events did not include stable angina, intermittent claudication, or chronic ischemic foot ulcers or gangrenes?

#8 Severe comorbidities

The authors determined severe comorbidities as chronic pulmonary disease, chronic liver disease and/or cancer. What was the definition of each disorder? Chronic liver disease included fatty liver and NAFLD? Cancer included that treated at the earliest stage years ago? The reviewer is afraid that the diseases might not always reflect “severe” comorbidities which would affect the determination of individualized HbA1c goals.

In addition, the reviewer is also afraid that the current population might include patients with liver cirrhosis or those currently treated with cancer. HbA1c levels in these patients would be falsely reduced; consequently, the association of lower HbA1c levels with mortality risk would be biased.

#9 “Frailty” scoring

The authors determined “personalized HbA1c goals” objectively based on patients’ characteristics from the viewpoint of “frailty”. The idea was very interesting. However, was the algorism validated? If so, please cite the references. If not so, the current study should be positioned as the study to validate this tentatively developed algorism, or to develop a more sophisticated one. The current findings were considerably dependent on this algorism, and therefore its validation was quite important.

The notion of “personalized HbA1c goals” is very important in clinical practice, but at the same time makes it difficult to assess the achievement of goals in clinical studies. If the authors successfully developed an objective and valid algorism, the algorism would become a base of future studies, like the Charlson risk index.

The reviewer additionally recommends not to call the related factors was “frailty” factors, because frailty is not clinically determined by a composite of age, duration of diabetes, complications and comorbidities. Calling these factors as “frailty” factors is misleading.

#10 HbA1c target

The current study was important because the study enrolled patients between 2006 and 2008, when the recommended HbA1c target was <7.0% for all patients, as the authors emphasized. Consequently, the study enabled to analyze the population to whom “a uniformly recommended goal”, but not “personalized goals”, was set. This was a major strength of the current study. However, the authors included the follow-up data even after “personalized goals” were recommended. To keep the strength of the current study, the data after the recommendation of “personalized goals” should be excluded. “Personalized goals” have been officially recommended since 2012, when the position statement of the ADA and the EASD was published by Inzucchi, et al. The data after the statement should be excluded.

#11 Unnecessary margin of HbA1c by 0.5%

The authors added 0.5% to HbA1c goals, but this margin of 0.5% was uncalled-for. When the HbA1c goal is set to be <7.0%, then the achievement of the goal is HbA1c < 7.0%, not <7.5%. HbA1c higher than 7.0% is not the achievement.

#12 Risk of multicollinearity

Waist circumference was estimated from BMI. The co-adjustment for the two variables in the regression model would cause multicollinearity. Similarly, the co-adjustment for systolic and diastolic blood pressure would cause multicollinearity.

#13 Adjustment for lipid profiles

Why did the authors adjust for triglycerides, total and HDL cholesterol, although they had the data on non-HDL cholesterol and LDL cholesterol? The findings were unchanged when non-HDL cholesterol levels, or alternatively LDL cholesterol levels, were used for adjustment?

#14 HbA1c measurement under treatment with medications

Were HbA1c levels measured with medication regimens unchanged for months? If HbA1c levels did not reflect the treatment effect of medications, the interpretation of the findings would be difficult.

#15 Figure 1 and 2

The authors illustrated the estimated overall survival rate using the Cox proportional hazards regression model. However, were the hazards really proportional?

#16 Previous reports from the same registry

There were several previous reports from the same registry, i.e., the RIACE Study. For example, the study group already revealed that the past HbA1c variability was a strong predictor of mortality in about a half of the study population (Diabetes Obes Metab 2018, 20, 1885-1993). The medication regimens and the achievement of HbA1c goals would be reasonably influenced by the information of the past HbA1c variability in clinical practice. Please check whether the current findings were unchanged even after further adjustment for the past HbA1c variability in this part of the study population. In addition, another article of their study group (Cardiovascular Diabetology 2013, 12, 98) reported that the association of HbA1c variability was not the same among CVDs: coronary or cerebrovascular event, myocardial infarction, stroke, any lower limb vascular event, and ulceration/gangrene. These findings indicated that phenotypes of CVDs would be good markers of the past HbA1c variability. So please check whether the current findings were unchanged when these phenotypes of CVDs, instead of a composite of any CVD, were used as covariates in the whole study population.

Another study, analyzing the prognostic impact of DKD (Acta Diabetologica 2018, 55, 603-612), demonstrated that the prognostic impact was different among eGFR 45-60, 30-45, and <30, and between microalbuminuria and macroalbuminuria. In addition, the RECPAM analysis suggested that the prognostic impact of a covariate was different between subgroups divided according to another covariate; there seemed to be interaction effects. However, the current study did not take into account these aspects. Please check whether the current findings were unchanged even after considering these aspects.
They also revealed that gender differences in treatments (J Intern Med 2013, 274, 176-191). To address the issue on therapeutic targets, these aspects should be additionally included in the current analysis.
Furthermore, it is also important to show whether their previous findings were unchanged even after further adjustment for the covariates considered in the current study. For example, in their previous studies, the prognostic impact of the past HbA1c variability (Diabetes Obes Metab 2018, 20, 1885-1993) and DKD (Acta Diabetologica 2018, 55, 603-612, and Diabetologia 2018, 61, 2277-2289) were not adjusted for HbA1c goals, anti-diabetic treatment, or severe comorbidities other than cancer. The study group have responsibility to clarify whether their impacts were unchanged (i.e., still significant) even after further adjustment for these factors.

#17 High mortality risk in the current study population

The mortality risk in the current study population seemed somewhat higher. The 10-year cumulative incidence rate of all-cause mortality looked higher than 20%. Was this study population a representative of a general population with type 2 diabetes? Or were there any cautions in generalizing the current findings?

#18 Statistical power

The current database was derived from a very large clinical study, but they divided the study population to somewhat many subgroups. Each subgroup (especially those without SU, glinide, or insulin) was not always large. In addition, the statistical power depends on the number of events, rather than the number at risk. The number of adjusting factors also attenuate the statistical power. The reviewer is afraid that the number of events (deaths) might not be so large enough (especially in those without SU, glinide, or insulin). Did the authors check the statistical power? Insufficient statistical power causes the difficulty in interpreting non-significant associations.

#19 Suggestion

Taken together, it would not be adequate to evaluate the intervention effect of certain HbA1c levels under certain medications, using the current database. Furthermore, the study group have responsibility to clarify the association between the current manuscript with their previous reports. The reviewer appreciates that their database was obtained from a very large prospective study, and therefore that the findings from their database would greatly contribute to the management of type 2 diabetes in clinical practice. The reviewer would like the authors to publish their data, with appropriate statistical approaches and interpretations.

Reviewer 3 Report

This is a potentially important observational study of the impact of glycaemic control levels on subsequent mortality.  The authors did something unusual and interesting in that they attempted to assign appropriate glycaemic targets using available data to their study group then they examined mortality outcomes by both actual and assigned glycaemic targets. The cohort is large, there is quite a deal of descriptive data and the follow-up period is long. Consequently the results are potentially of considerable interest to a wide range of clinicians who treat patients with diabetes. However I have two major concerns, the first is conceptual and the second relates to the data presentation.

This is not a frailty study and the use of the term should probably be removed from the paper. There is now considerable research on what frailty might be but it cannot be defined by their chosen methods. With their dataset, they could probably use the Rockwood deficit accumulation method to define frailty at least in those aged over 65 years (I believe that is the cut-off age).  Using age, duration, CHD events and comorbidities may be justifiable but I doubt it. The altered glycaemic targets would seem a very crude method. Perhaps simply looking at macro and micro vascular complications alone as target modifiers would be simpler given the RCT results. The data presentation is extremely difficult to follow and renders the paper tedious. The survival curves are hard to discern and I believe one or more tables would serve better.

Other issues:

What were the inclusion/exclusion criteria for RIACE? The Introduction provides a lengthy discussion on the few RCTs yet much of concerns around targets come from observational studies. Perhaps this section could be broadened and also shortened. Too much info provided on the renal measures in Methods. Why only explore the impact of hypoglycemic drugs? There is at least one recent study showing that metformin may be associated with mortality in older patients (Diabet Obesity Metab 2018:20:2852). 2 decimal places suffice for HRs etc.